# Peptides Isolated from Amphibian Skin Secretions with Emphasis on Antimicrobial Peptides

**DOI:** 10.3390/toxins14100722

**Published:** 2022-10-21

**Authors:** Xi Chen, Songcai Liu, Jiayuan Fang, Shuo Zheng, Zhaoguo Wang, Yingying Jiao, Peijun Xia, Hongyan Wu, Ze Ma, Linlin Hao

**Affiliations:** College of Animal Science, Jilin University, 5333 Xi’an Road, Changchun 130062, China

**Keywords:** amphibian skin secretions, peptides, origin, primary structure, functions

## Abstract

The skin of amphibians is a tissue with biological functions, such as defense, respiration, and excretion. In recent years, researchers have discovered a large number of peptides in the skin secretions of amphibians, including antimicrobial peptides, antioxidant peptides, bradykinins, insulin-releasing peptides, and other peptides. This review focuses on the origin, primary structure, secondary structure, length, and functions of peptides secreted from amphibians’ skin. We hope that this review will provide further information and promote the further study of amphibian skin secretions, in order to provide reference for expanding the research and application of amphibian bioactive peptides.

## 1. Introduction

A myriad of peptides from animals and plants have been documented. These peptides hold potential physiological functions in humans, mainly including antimicrobial [1], antioxidative [2,3], antithrombotic [4,5], and antihypertensive [6,7], depending on their structural properties. To date, peptides are isolated and identified from legumes [8], cereal [9], fish-derived products [10,11], porcine skin [12,13], antler [14,15], and amphibians skin, among which, the most studied peptides are the ones found in the amphibians skin, due to the amphibians skin’s unique chemical properties [16].

Amphibians skin is directly exposed to a variety of environments, without scales and hair, and it not only protects amphibians from the effects of the external environment, but also performs various functions, such as respiration, osmoregulation, and thermoregulation [17]. The peptides, stored in amphibian skin granular glands, can be released in high concentrations into skin secretions when the amphibians are stressed or injured. Peptides, including antimicrobial peptides [18,19], antioxidant peptides [20,21], antiviral peptides [22,23], antitumor peptides [24,25], and other peptides [16], are extracted, as regulating body internal functions and fertility of traditional Chinese medicine, as well as the ancient Egyptian medicine to cure the pain and diarrhea [26,27,28,29].

In this paper, exploration of the structures and biological functions of peptides from the skin secretions of amphibians were described, which provides theoretical reference for expanding the research field of peptides.

## 2. Antimicrobial Peptides

Antimicrobial peptides (AMPs), an important part in innate immune defense of amphibians [30], have been reported to have an excellent broad-spectrum antibacterial function and low biological toxicity [31]. AMPs sterilize and inhibit bacteria by interacting with bacterial cell membrane [32]. AMPs can selectively combine with the outer membrane of bacteria to form cavities in the cell membrane, which leads to the outflow of nutrients, internal ions, other cellular contents, and bacterial death. As we all know, alpha helix (α-helix) is necessary for AMPs to resist bacteria [33]. There are three main membrane permeation mechanisms of α-helix AMPs: Barrel-stave mechanism [34], Carpet model [35], and Toroidal pore model [36] (Figure 1). They are used to explain the AMPs action on bacterial membrane, but the mechanism is not clear.

Esculentins, brevinins, ranatuerins, ranacyclins, temporins, bombinins, and dybowskins are the most famous and representative amphoteric cationic AMPs in amphibian skin secretions [37].

**Figure 1 toxins-14-00722-f001:**
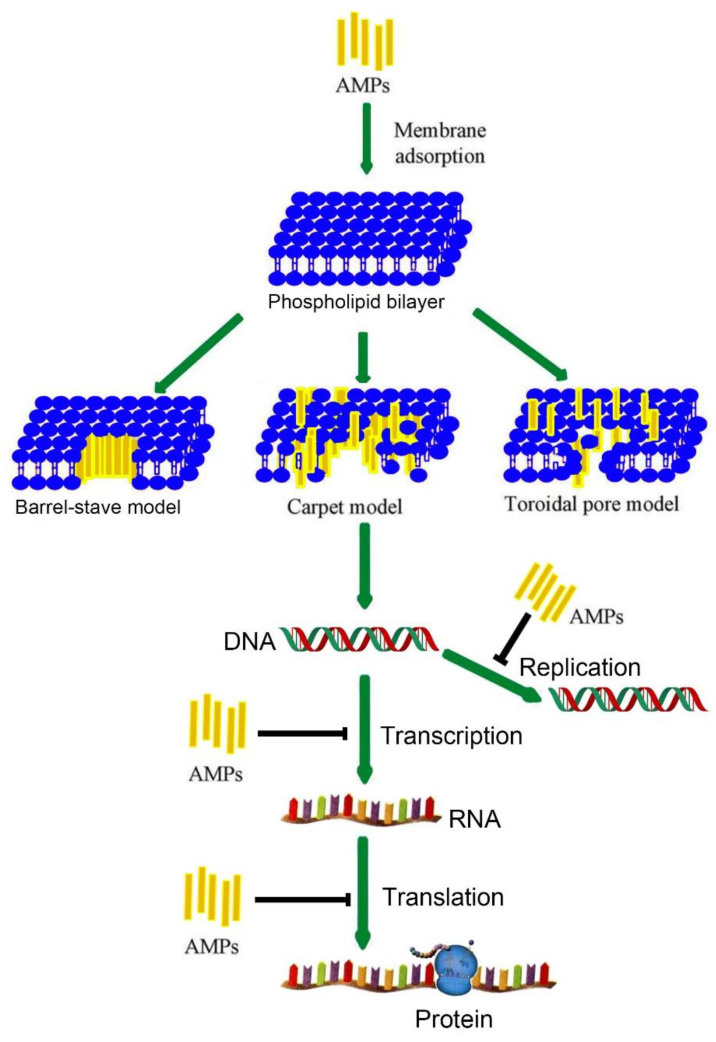
Mode of action for AMPs activity [36,38]; “⊥” indicates an inhibitory effect. Barrel-stave model: Under the action of electrostatic attraction, positively charged AMPs aggregate to negatively charged bacterial membrane surface. AMPs induce and participate in continuous bending of monolayer phospholipid membrane. AMPs are inserted into the water channel composed of phospholipid head groups through small holes [39]. Carpet model: When the AMPs are spread to the surface of the membrane until the concentration of the domain is reached, small holes will be formed on the membrane surface, and then the membrane will be destroyed, leading to cell death [40]. Toroidal pore model: AMPs are arranged perpendicular to the cell membrane in the plasma membrane, and their hydrophobic sides are outside. The polar surface forms holes, through which the contents of bacterial cytoplasm are lost, leading to cell death [36]. AMPs can inhibit DNA, RNA, and protein synthesis.

### 2.1. Esculentins

Esculentins (Table 1) were isolated from the skin secretions of North American and Asian frogs, which contain esculentin-1 and esculentin-2. Both esculentin-1 and esculentin-2 are circular peptides, with two conserved Cys residues forming an intramolecular disulfide bond at the C-terminus of the molecule, resulting in a ring structure of seven peptides at the C-terminus.

Esculentin-1 consists of 46 amino acids. Esculentin-1 family has a well-conserved primary structure with only a handful of similar polar amino acid residue substitutions. The ring domain is highly conserved among the cationic and seven amino acids. Esculentin-1 has strong inhibitory activity against a variety of pathogenic bacteria (minimum inhibitory concentration (MIC) < 1 μM), such as *E. coli*, *S. aureus*, *P. aeruginosa*, and *C. albicans* [41,42].

Esculentin-2 is slightly shorter, typically containing 37 amino acids [41]. Members of the esculentin-2 family have more amino acid substitutions than esculentin-1, mostly in amino acids of similar properties. Esculentin-2 shows different antimicrobial activity against *E. coli* (MIC < 10 μM), *S. aureus* (MIC < 10 μM), and *C. albicans* (MIC, 30–50 μM) [42].

**Table 1 toxins-14-00722-t001:** The origin, primary structure, secondary structure, length, and antibacterial activity of esculentins. Secondary structure prediction was performed by GOR IV algorithm (https://npsa-prabi.ibcp.fr/cgi-bin/npsa_automat.pl?page=/NPSA/npsa_gor4.html accessed on 27 September 2022); Hh: Alpha helix; Ee: Extended strand; Cc: Random coil; MIC: Minimum inhibitory concentrations (μM); NA: not active.

Peptides Name	Species	Primary Structure	Secondary Structure	Length	MIC (μM)	Ref.
Esculentin-1	*Rana esculenta*	GIFSKLGRKKIKNLLISGLKNVGKEVGMDVVRTGIDIAGCKIKGEC	Hh: 19.57% Ee: 39.13% Cc: 41.30%	46	0.2 (*E. coli* D21) 0.1 (*B. megaterium* BmII) 0.4 (*S. aureus* Cowan 1) 0.7 (*P. aeruginosab* ATCC15692) 0.5 (*C. Albican*) 0.9 (*S. Cerevisiae*)	[43]
Esculentin-1SEa	*Rana sevosa*	GLFSKFNKKKIKSGLIKIIKTAGKEAGLEALRTGIDVIGCKIKGEC	Hh: 36.96% Ee: 26.09% Cc: 36.96%	46	1.1 (*E. coli*) 1.21 (*M. luteus*)	[44]
Esculentin-2a	*Rana ridibunda*	GILSLVKGVAKLAGKGLAKEGGKFGLELIACKIAKQC	Hh: 48.65% Ee: 8.11% Cc: 43.24%	37	0.4 (*E. coli* D21) 0.2 (*B. megaterium* BmII) 0.8 (*S. aureus* Cowan 1) 3.5 (*P. aeruginosab* ATCC15692)	[44]
Esculentin-2SE	*Rana sevosa*	GFFSLIKGVAKIATKGLAKNLGKMGLDLVGCKISKEC	Hh: 43.24% Ee: 32.43% Cc: 24.32%	37	0.3 (*E. coli*) 0.36 (*M. luteus*)	[44]
Esculentin-2-ALa	*Amolops loloensis*	GIFALIKTAAKFVGKNLLKQAGKAGLEHLACKANNQC	Hh: 62.16% Ee: 13.51% Cc: 24.32%	37	12.5 (*E. coli*) 7.5 (*B. dysenteriae*) 25 (*A. calcoaceticus*) 50 (*P. aeruginosa*) 2.5 (*S. aureus*) 2.5 (*B. pumilus*) 7.5 (*B. cereus*)	[45]
Esculentin-2-ALb		GIFSLIKTAAKFVGKNLLKQAGKAGVEHLACKANNQC	Hh: 56.76% Ee: 5.41% Cc: 37.84%	37	12.5 (*E. coli*) 7.5 (*B. dysenteriae*) 12.5 (*A. calcoaceticus*) 50 (*P. aeruginosa*) 1.25 (*S. aureus*) 2.5 (*B. pumilus*) 7.5 (*B. cereus*)	
Esculentin-2CHa	*Lithobates. chiricahuensis*	GFSSIFRGVAKFASKGLGK DLAKLGVDLVA CKISKQC	Hh: 54.05% Cc: 45.95%	37	<6 (*S. aureus*)	[46]
Esculentin 2EM	*Glandirana emeljanovi*	GILDTLKQFAKGVGKDLVKGAAQGVLSTVSCKLAKTC	Hh: 27.03% Ee: 32.43% Cc: 40.54%	37	>75 (*E. coli*) <6.25 (*S. aureus*)	[47]

### 2.2. Brevinins

Brevinins (Table 2) consist of two kinds of antibacterial peptides: Brevinin-1, which usually consists of 24 amino acids, and brevinin-2, which contains 33 amino acids. Morikawa et al. [48] isolated the first members of brevinins from the skin secretions of *Rana brevipoda porsa* in 1992 and named them brevinin-1 and brevinin-2, respectively.

The primary and secondary structures of brevinin-1 family peptides are relatively conserved, and the 14th position of the N-terminal peptide is mostly Pro residue. Structure- and function-related studies suggested that this residue generates a stable conjugation in the molecule and may play a significant role in transmembrane pore formation [49]. The C-terminal of brevinin-1 is composed of conserved seven amino acid sequences: Cys-3X-Lys/Arg-Lys-Cys, which forms an intramolecular disulfide bond between the two cysteines, allowing it to form a conserved ring structure of seven amino acids at the C-terminal, named “Rana-box”. Brevinin-1 has strong anti-bacterial and anti-fungal activities as well as a strong hemolytic activity (HC_50_). In addition, brevinin-1 from *Rana japonicum* has anti-herpes virus type I and type II activities [50,51,52]. The obtained brevinin-1E from *Rana esculenta linneaus* had a hemolytic activity with less than 1 μM [50], which would limit its prospects for therapeutic use. Kwon et al. [53] reported that the linear acetamide cysteine-methylated brevinin-1 analogue could decrease the hemolytic activity while it did not have an impact on antibacterial activity, and brevinin-1 still had antiviral activity after reduction and carboxyamine methylation. Kumari et al. [54] showed that transferring the C-terminal domain of the heptapeptide ring to the middle region of the peptide chain could also decrease the hemolytic activity without impairing its antimicrobial activity. In addition, these findings suggested that brevinin-1 can be used as a potential new drug target.

Brevinin-2 family peptides consist of 33 or 37 amino acid residues, the first amino acid in the N-terminal is Gly, the C-terminal has “Rana-box”, the isoelectric point is greater than 7, carries 1–5 positive charges, contains α-helical structure, and is a typical frog antimicrobial peptide. Brevinin-2 obtained from *Rana esculenta* [55,56] and *Rana ornitiventrisprice* [57] has a strong antibacterial activity against *E. coli*, *S. aureus,* and *C. albicans*. The hydrophilic and hydrophobic amino acids of this type of AMPs are alternately distributed, with typical amphiphilic properties, and the overall antibacterial activity is relatively high, which is suitable for modification.

**Table 2 toxins-14-00722-t002:** The origin, primary structure, secondary structure, length, and antibacterial activity of brevinins. Secondary structure prediction was performed by GOR IV algorithm (https://npsa-prabi.ibcp.fr/cgi-bin/npsa_automat.pl?page=/NPSA/npsa_gor4.html accessed on 27 September 2022); Hh: Alpha helix; Ee: Extended strand; Cc: Random coil; MIC: Minimum inhibitory concentrations (μM); NA: not active.

Peptides Name	Species	Primary Structure	Secondary Structure	Length	MIC (μM)	Ref.
Brevinin-1	*Rana brevipoda porsa*	FLPVLAGIAAKVVPALFCKITKKC	Hh: 33.33% Ee: 16.67% Cc: 50.00%	24	34 (*E. coli*) 8 (*S. aureus*)	[48]
Brevinin-1GHd	*Hylarana guentheri*	FLGALFKVASKLVPAAICSISKKC	Hh: 16.67% Ee: 37.50% Cc: 45.83%	24	8 (*E. coli*) 32 (*P. aeruginosa*) 2 (*S. aureus*) 4 (*MRSA*) 4 (*C. albicans*)	[58]
Brevinin-1AUa	*Rana aurora aurora*	FLPILAGLAAKLVPKVFCSITKKC	Hh: 12.50% Ee: 33.33% Cc: 54.17%	24	13 (*E. coli*) 25 (*P. aeruginosa*) 13 (*E. cloacae*) 50 (*K. pneumoniae*) 3 (*S. aureus*) 6 (*S. epidermidis*) 3 (*C. albicans*)	[59]
Brevinin-1AUb		FLPILAGLAANILPKVFCSITKKC	Ee: 33.33% Cc: 66.67%	24	25 (*E. coli*) 25 (*P. aeruginosa*) 25 (*E. cloacae*) >50 (*K. pneumoniae*) 3 (*S. aureus*) 6 (*S. epidermidis*) 3 (*C. albicans*)	
Brevinin-1BYa	*Rana boylii*	FLPILASLAAKFGPKLFCLVTKKC	Hh: 16.67% Ee: 33.33% Cc: 50.00%	24	17 (*E. coli*) 2 (*S. aureus*) 3 (*C. albicans*)	[50]
Brevinin-1BYb		LPILASLAAKLGPKLFCLVTKKC	Hh: 13.04% Ee: 30.43% Cc: 56.52%	23	16 (*E. coli*) 4 (*S. aureus*) 16 (*C. albicans*)	
Brevinin-1BYc		LPILASLAATLGPKLLCLITKKC	Ee: 39.13% Cc: 60.87%	23	NA (*E. coli*) 8 (*S. aureus*) 35 (*C. albicans*)	
Brevinin-1SY	*Rana sylvatica*	FLPVVAGLAAKVLPSIICAVTKKC	Ee: 41.67% Cc: 58.33%	24	45 (*E. coli*) 7 (*S. aureus*)	[51]
Brevinin-1LTa	*Hylarana latouchii*	FFGTALKIAANVLPTAICKILKKC	Hh: 25.00% Ee: 29.17% Cc: 45.83%	24	80 (*P. fluorescens*) 10 (*S. aureus*) 6 (*B. subtilis*) 50 (*C. albicans*)	[60]
Brevinin-1Sa	*Rana sphenocephala*	FLPAIVGAAGQFLPKIFCAISKKC	Ee: 41.67% Cc: 58.33%	24	55 (*E. coli*)	[61]
Brevinin-1Sb		FLPAIVGAAGKFLPKIFCAISKKC	Ee: 54.17% Cc: 45.83%	24	17 (*E. coli*)	
Brevinin-1Sc		FFPIVAGVAGQVLKKIYCTISKKC	Ee: 62.50% Cc: 37.50%	24	14 (*E. coli*)	
Brevinin-2	*Rana brevipoda porsa*	GLLDSLKGFAATAGKGVLQSLLSTASCKLAKTC	Hh: 39.39% Ee: 6.06% Cc: 54.55%	33	4 (*E. coli*) 8 (*S. aureus*)	[48]
Brevinin-2E	*Rana esculenta*	GIMDTLKNLAKTAGKGALQSLLNKASCKLSGQC	Hh: 48.48% Cc: 51.52%	33	0.5 (*E. coli* D21) 0.2 (*B. megaterium* BmII) 2 (*S. aureus* Cowan 1) >30 (*P. aeruginosab* ATCC15692) NA (*C. albican*) NA (*S. cerevisiae*)	[43]
Brevinin-2Oa	*Rana ornativentris*	GLFNVFKGALKTAGKHVAGSLLNQLKCKVSGGC	Hh: 42.42% Ee: 27.27% Cc: 30.30%	33	2 (*E. coli*) 10 (*S. aureus*) 40 (*C. albicans*)	[62]
Brevinin-2Ob		GIFNVFKGALKTAGKHVAGSLLNQLKCKVSGEC	Hh: 45.45% Ee: 33.33% Cc: 21.21%	33	4 (*E. coli*) 9 (*S. aureus*) 40 (*C. albicans*)	

### 2.3. Ranatuerins

In their study, Goraya et al. [63] isolated ranatuerins (Table 3) from the skin secretions of *Rana catesbeiana*. There are 25 amino acid residues in ranatuerin-1, and the C-terminus of the molecule contains a circular domain consisting of seven amino acids. Ranatuerins family has a well-conserved primary structure with only individual amino acid substitutions. Ranatuerin-1 has a strong antimicrobial activity and can inhibit *S. aureus*, *E. coli*, and *C. albicans* [63], and was considered as an ideal substitute for antibiotics. The structure of ranatuerin-1 consists of an α-helix, β-lamellar, and β-turn [64]. Substituting Lys residues for Gly residues at positions 10, 13, and 15 to destroy β-lamellar, which can significantly decrease the antibacterial activity of ranatuerin-1, suggest that the β-lamellar structure plays a key role in the antibacterial activity of ranatuerin-1.

The ranatuerin-2 family members have amino acid residues ranging from 28 to 31, and the primary structure of ranatuerin-2 family members varies greatly. Ranatuerin-2 shows high bacteriostatic activity against Gram-positive and Gram-negative microbes and has a low hemolytic activity against human erythrocytes [59]. The MIC of ranatuerin-2 against *E. coli* ranges from 2 to 30 μM and against *S. aureus* from 2 to >200 μM. Ranatuerin-2 has a low antimicrobial activity against *C. albicans* (MIC = 35 μM). The HC_50_ values of ranatuerin-2 against human erythrocytes are in the range of 35 to >200 μM [42].

### 2.4. Ranacyclins

Ranacyclins (Table 4) were isolated from the skin secretions of *Rana temporaria* and *Rana esculenta*, respectively [70]. In addition, there is a peptide leucine arginine (pLR) homologous loop region [70]. A hallmark of the ranacyclins is the nearly entirely conserved region of 13 amino acids that form a cyclic undecapeptide through a disulphide bridge connecting Cys^5^ and Cys^15^ [41]. Ranacyclins interact primarily with the hydrophobic core of the cell membrane, not with the negatively charged lipid head group. Therefore, ranacyclins can combine and intercalate into both zwitterionic and negatively charged membrane vesicles [16,71].

Ranacyclin E and ranacyclin T were found in the skin secretions of *Rana temporaria* and *Rana esculenta* [70]. Ranacyclin E has potent antibacterial activity against *S. lentus* and *P. syringae pv tabaci* [16,70]. The antibacterial activity of ranacyclin T is similar to ranacyclin E [70,71]. Ranacyclin-B-RN1 and ranacyclin-B-RN2 from *H. nigrovittata* show an inhibitory effect on the growth of *S. aureus* at concentrations of up to 6 and 12.7 μM [16].

### 2.5. Temporins

Temporins (Table 5) were initially identified from the skin secretions of *Rana esculenta* [73] and *Rana temporaria Linnaeus* from Europe [74]. Temporins were also isolated from *Rana pipiens* [52,75], *Rana esculenta* [55], *Rana grylio* [76], *Rana tsushimensis* [77], and *Hylarana guentheri* [78]. This is the smallest α-helical amphipathic AMP that occurs in nature. The peptide molecule consists of 10–14 amino acid residues and contains one basic amino acid residue (Lys or Arg). In hydrophobic environments, temporins fold into an amphipathic α-helix, and have cationic properties with pH from 2 to 3 [79]. The molecular diversity of temporins peptides determines their functional diversity. Some temporin peptides, such as temporin L have diverse biological activities against bacteria, viruses, fungi, and protozoa, with MICs ranging from 1 to over 100 μM; other temporin proteases exhibit immunomodulatory, transplant infection, anticonvulsant, killing tumor cells, and other physiological activities. Structural and activity studies of temporin A as well as the substitution of Ile residues with Leu can increase the antibacterial activity [80]. Temporin-1Ta and temporin-1Tb are widely used due to their low hemolytic activity [81]. Given that the outer lipopolysaccharide (LPS) can inactivate temporin-1Ta, temporin-1Tb, and other AMPs, a six-residue aromatic cationic peptide WKRKRF named β-boomerang motif folds into a compact “boomerang” structure after interacting with LPS and is incorporated into the temporal protein [82]. Therefore, the inclusion of this motif in AMPs may effectively eliminate outer LPS membrane-induced aggregation, producing broad-spectrum activity. Temporin-1Tl is active against Gram-positive and Gram-negative bacteria [83,84], but it has high hemolytic and cytotoxic activities and low therapeutic index [85]. In the presence of sodium dodecyl sulfate (SDS) and dodecylphosphocholine (DPC) micelles, temporin-1Ta and temporin-1Tl were analyzed by spectroscopic techniques (CD and NMR) and molecular dynamics simulation, simulating the negatively charged membrane and zwitterionic membrane, respectively. Peptides are located at the micelle-water interface in SDS, and they tend to be perpendicular to the micellar surface, with the N-terminal embedded in the hydrophobic core in DPC. However, differences between the two peptides are present: Compared with temporin-1Ta, temporin-1Tl has higher trend in both membrane simulation systems to form an α- helical structure and penetrate lipid vesicles [86]. Studies of temporins interacting with liposomes composed of different phospholipids showed that its antibacterial mechanism is via transmembrane pores formation, which gives rise to bacterial death in the end [87]. To study the bactericidal action of temporins, we hypothesize that we can use a model named “barrel-stave model” (Figure 1), where AMPs are electrostatically attached to the surface of negatively charged cell membranes and then inserted directly into the membrane to form transmembrane voids.

### 2.6. Bombinins

The first identified amphibian skin AMP was bombinin, which originated from *Bombina variegate* [90]. The family peptides can be divided into bombinins, bombinin-like peptides (BLPs), and bombinins H (Table 6). The BLPs exhibit substantial homology with other AMPs. BLP-1, BLP-2, and BLP-3 were found in the skin secretions of *Bombina orientalis*, respectively [91]. Analyzing the nucleotide sequences of the bombinins precursors led to the discovery of bombinin H. In addition, bombinin H was isolated from the skin secretions of *B. maxima* and *B. orientalis* [92,93]. Bombinin H consisted of 17 or 20 amino acids. In addition to their equivalent L-isomers, bombinin H contains several peptides with a second-ranked D-amino acid [94].

Bombinins are effective against Gram-positive and Gram-negative bacteria, as well as fungi, but are almost ineffective in hemolysis tests [95]. Xiang et al. [96] revealed that BHL-bombinin is active against *S. aureus* (MIC = 1.6 μM) and *E. coli* (MIC = 6.6 μM). Peng et al. [97] reported that the MIC of bombinin-BO1 against *S. aureus* and *E. coli* are 26.3 and 26.3 μM, respectively. However, there is a low activity of bombinin H against bacteria, but it lyses erythrocytes. Peng et al. [97] revealed that bombinin H-BO1 is active against *S. aureus* (MIC > 161.1 μM) and *E. coli* (MIC > 161.1 μM). Xiang et al. [96] revealed that the MIC of bombinin HL against *S. aureus* is 156.8 μM, while it is not active against *E. coli*.

**Table 6 toxins-14-00722-t006:** The origin, primary structure, secondary structure, length, and antibacterial activity of bombinins. Secondary structure prediction was performed by GOR IV algorithm (https://npsa-prabi.ibcp.fr/cgi-bin/npsa_automat.pl?page=/NPSA/npsa_gor4.html accessed on 27 September 2022); Hh: Alpha helix; Ee: Extended strand; Cc: Random coil; MIC: Minimum inhibitory concentrations (μM); NA: not active.

Peptides Name	Species	Primary Structure	Secondary Structure	Length	MIC (μM)	Ref.
BHL-bombinin	*Bombina orientalis*	GIGGALLSFGKSALKGLAKGLAEHF	Hh: 44.00% Ee: 24.00% Cc: 32.00%	25	6.6 (*E. coli*) 26.2 (*P. aeruginosa*) 1.6 (*S. aureus*) 6.6 (*MRSA*)	[96]
Bombinin HL		LLGPVLGLVSNVLGGLL	Ee: 58.82% Cc: 41.18%	17	NA (*E. coli*) NA (*P. aeruginosa*) 156.8 (*S. aureus*) NA (*MRSA*)	
Bombinin-BO1	*Bombina orientalis*	GIGSAILSAGKSIIKGLAKGLAEHF	Hh: 28.00% Ee: 20.00% Cc: 52.00%	25	26.3 (*E. coli*) 26.3 (*S. aureus*) 52.5 (*C. albicans*)	[97]
Bombinin H-BO1		IIGPVLGLVGKALGGLL	Ee: 47.06% Cc: 52.94%	17	26.3 (*E. coli*) 26.3 (*S. aureus*) 52.5 (*C. albicans*)	
Bombinin H1	*Bombina variegata*	IIGPVLGMVGSALGGLLKKIG	Hh: 23.81% Ee: 38.10% Cc: 38.10%	21	3.8 (*E. coli* D 21) 2.1 (*S. aureus* Cowan 1)	[98]
Bombinin H3		IIGPVLGMVGSALGGLLKKIG	Hh: 23.81% Ee: 38.10% Cc: 38.10%	21	3.7 (*E. coli* D 21) 2.4 (*S. aureus* Cowan 1)	
Bombinin H4		LIGPVLGLVGSALGGLLKKIG	Hh: 23.81% Ee: 42.86% Cc: 33.33%	21	4.8 (*E. coli* D 21) 3.3 (*S. aureus* Cowan 1)	

### 2.7. Dybowskins

Dybowskins (Table 7) were identified in the skin secretions of *Rana dybowskii*. Kim et al. [99] found that dybowskin-1 and dybowskin-2 were both isomors, and the difference lies in the two amino acid residues at the 7th and 14th positions of the N-terminal. From dybowskin-3 to dybowskin-6, they both differed in size and sequence. All the dybowskins showed a strong antimicrobial activity against the Gram-positive and Gram-negative bacteria (MIC, from 12.5 to >100 μM), as well as the fungus (MIC, from 25 to >100 μM) [99]. Jin et al. [100] reported that dybowskin-1CDYa, dybowskin-2 CDYa, and dybowskin-2CDYb had different amino acid compositions and little similarity to the known AMPs sequence. The mature dybowskin-1CDYa and dybowskin-2CDYa had a strong antimicrobial activity and little effect on the hemolysis of human erythrocytes. The dybowskin-1CDYa consisted of 13 amino acids and the dybowskin-2CDYa consisted of 18 amino acids. Yang et al. [69] found that the amino sequences of dybowskin-YJa and dybowskin-YJb were less similar to other antimicrobial peptides isolated from the Rana species. Both dybowskin-YJa and dybowskin-YJb were ineffective against all strains.

## 3. Antioxidant Peptides

Antioxidant peptides (Table 8) are an important area of scientific interest, which have the important ability to clear free radicals in the body, maintain the normal function of organelles, and keep the body stable. The mechanism of action of antioxidant peptides is a new defense mechanism named “third antioxidant system” [101]. At present, antioxidant peptides have been extracted and isolated from the skin secretions of amphibians. Some researchers have extracted and isolated a variety of antioxidant peptides from the skin secretions of *Rana. plyuraden* and analyzed their primary structures. It was found that these antioxidant peptides share highly homologous preproregions, although there were significant differences in these mature peptide regions, suggesting that these antioxidant peptides may come from a common ancestor [101]. In addition, a variety of antioxidant peptides from different families have been found in the skin secretions of *Odorrana livida**, Odorrana schmackeri,* and *Odorrana andersonii.* Liu et al. [102] reported that antioxidin-RL was identified from the skin secretions of *Odorrana livida*. It eliminates most of the 2, 2′-azino-bis (3-ethylbenzthiazoline-6-sulfonic acid) free radical in 2 s, significantly faster than the commercial antioxidant factor butylated hydroxytoluene, suggesting a potentially significant impact on redox homeostasis in amphibians’ skin. Xie et al. [103] identified a new antioxidant peptide (named OS-LL11) from the skin secretions of *Odorrana schmackeri*. OS-LL11 directly scavenged by reducing levels of catalase, Keap-1, HO-1, GCLM, and NQO1, maintaining the viability of mouse keratinocytes in mice exposed to ultraviolet B (UVB) or hydrogen peroxide (H_2_O_2_). Yin et al. [104] reported that a short gene-coding peptide (OA-VI12) was identified from the skin secretions of *Odorrana andersonii.* OA-VI12 protected cell viability, promoted catalase release, and reduced the levels of lactic dehydrogenase and reactive oxygen species (ROS).

## 4. Bradykinin Peptides

Bradykinin is mainly a class of peptides in the kallikrein-bradykinin system, which eventually degrades to contain C-CO_2_H residue. Bradykinin plays its physiological roles through two different receptors β1 and β2. Among them, β2 is widely distributed in animals, and bradykinin mostly protects the heart through β2 [112]. At present, bradykinin has been found in the skin secretions of amphibians. For example, a novel bradykinin (bombinainin M) consisting of 19 amino acid residues has been found in the skin secretions of toad [113,114]. More than 10 bradykinins have been shown to be effective in expanding blood vessels, regulating blood pressure, and preventing cardiac insufficiency [42,75]. For example, Zhou et al. [115] reported that RR-18 displays an antagonism of bradykinin-induced rat ileum contraction and arterial smooth muscle relaxation. Arichi et al. [116] reported that bradykinin (BK) has negative contractile and variable properties to cardiac contractility, and BK regulates the excitability of intrinsic neurons in the heart by activating non-selective cationic channels and inhibiting M-type K channels through B^+^2 receptors. Compared with those in mammals, the biosynthetic pathway of bradykinin and its precursor biosynthase in amphibians need to be further explored.

## 5. Insulin-Releasing Peptides

Insulin-releasing peptides are types of active peptides, which can regulate insulin secretion in vivo and reduce blood glucose. In recent years, a variety of insulin-releasing peptides with different structures have been discovered and isolated from the skin secretions of amphibians (e.g., *Rana palustris* and *Bombina variegate*) [117]. For example, Manzo et al. [118] reported that pseudhymenochirin-1Pb and pseudhymenochirin-2Pa adopt a well-defined α-helical conformation. In the membrane-mimetic solvent, 50% (*v*/*v*) of trifluoroethanol-H_2_O extends over almost all the sequence and incorporates a flexible bend. Srinivasan et al. [119] reported that tigerinin-1R (RVCSAIPLPICH) enhances insulin release and improves glucose tolerance, suggesting that tigerinin-1R shows potential for development into novel therapeutic agents for treatment of type 2 diabetes mellitus. Abdel-Wahab et al. [120] reported that pseudin-2 is a cationic α-helical peptide. Pseudin-2 stimulates insulin secretion from BRIN-BD_11_ cells through a mechanism involving the Ca^2+^-independent pathway and identifies [Lys^18^]-pseudin-2 as a peptide with potential to develop valuable insulin drugs as a treatment for type 2 diabetes. In addition, some AMPs are not only antibacterial, but also have the activity of promoting insulin-releasing peptides, for example, the antimicrobial peptide plasticin-L1 obtained from the skin secretions of *Leptodactylus laticeps* not only has an antibacterial activity, but also has the activity of promoting insulin-releasing peptides [121].

## 6. Other Peptides

The skin secretions of amphibians (*Litoria* and *Uperoleia*) were found to be the main source of anticarcinogenic peptides. The anticancer peptides are mainly composed of α-helical aurein family and C-terminal aminated citropin family consisting of 16 amino acid residues. Both dermaseptin-PS1 isolated from *Phyllomedusa sauvagei* [122] and aurein isolated from *Litoria aureus* [123] are active peptides with strong anticancer activity.

A class of peptide compounds named bombesin were obtained from the skin secretions of *Bombina.* Since then, other peptides (Ranatensin, Litorin, and Phyllolitorin) similar in structure and function to bombesin have been extracted from other frog species. Bombesin acts on smooth muscle, constricts blood vessels, inhibits urine production, and has been shown to be effective in the treatment of neurological diseases [124,125].

In addition to the above species of peptides secreted by amphibians skin, other researchers have extracted bioactive substances, such as trypsin inhibitors [126], antihypotensive peptides [127], and neuropeptides [128,129,130,131]. The peptides from the skin secretions of amphibians are varied in structures and functions, which have played a very important role in promoting the evolution of natural organisms and the development of medical treatment.

## 7. Conclusions

Amphibians can live in different habitats and ecological environments due to the complex morphological characteristics of their skin, which has different functional bioactive substances. Many studies around the world indicated that amphibians may be the natural source of several drug candidates with antibacterial, antioxidant, antiviral, and anticancer properties. The growing problem of traditional antibiotic resistance and the urgent need for new antibiotics have aroused people’s interest in developing AMPs. First, different from traditional antibiotics, AMPs do not act on specific receptors on the bacterial cell membrane, but primarily through the interaction with the bacterial cell membrane, to achieve the purpose of bacteriostasis and sterilization. Therefore, the adverse consequences of bacterial resistance to AMPs through mutation does not easily appear. Second, based on the antibacterial mechanism of the interaction between AMPs and bacterial cell membranes, the positively charged AMPs have high affinity with LPS on the outer membrane of Gram-negative bacteria, replacing the divalent cations that bind with LPS to stabilize the membrane structure [132]. At the same time, LPS-binding ability of AMPs can prevent the occurrence of endotoxemia [133]. However, during the process of antibiotic-induced bacterial death, the production of proinflammatory cytokines caused by uncontrolled systemic LPS release will eventually lead to fatal septic shock. Although a variety of active peptides have been identified from the skin secretions of amphibians, and some of their biological activities have been known, their biological functions and mechanisms in amphibians need to be further explored. In addition, due to UV and the ecological environment destruction and other human factors, the influence of the amount and type of today’s global amphibians is falling. Therefore, we should try to study the molecular level of peptides in the synthesis of amphibians in the immune defense system and its regulatory mechanism, in order to provide a theoretical basis for its resource protection.

## Figures and Tables

**Table 3 toxins-14-00722-t003:** The origin, primary structure, secondary structure, length, and antibacterial activity of ranatuerins. Secondary structure prediction was performed by GOR IV algorithm (https://npsa-prabi.ibcp.fr/cgi-bin/npsa_automat.pl?page=/NPSA/npsa_gor4.html accessed on 27 September 2022); Hh: Alpha helix; Ee: Extended strand; Cc: Random coil; MIC: Minimum inhibitory concentrations (μM); NA: not active.

Peptides Name	Species	Primary Structure	Secondary Structure	Length	MIC (μM)	Ref.
Ranatuerin-1	*Rana catesbeiana*	SMLSVLKNLGKVGLGFVACKINKQC	Ee: 40.00% Cc: 30.30%	25	20 (*E. coli*) 20 (*P. aeruginosa*) 40 (*K. pneumoniae*) 20 (*E. cloacae*) >100 (*P. mirabilis*) 20 (*S. aureus*) 20 (*S. epidermidis*) 5 (*Streptococcus* Group B) 20 (*E. faecalis*)	[64]
Ranatuerin-1C	*Rana clamitans*	SMLSVLKNLGKVGLGLVACKINKQC	Ee: 28.00% Cc: 72.00%	25	1.5 (*E. coli*) 55 (*S. aureus*) 58 (*C. albicans*) 4 (*E. coli*) 17 (*S. aureus*) 14 (*C. albicans*)	[65]
Ranalexin-1Ca		FLGGLMKAFPALICAVTKKC	Ee: 50.00% Cc: 50.00%	20		
Ranatuerin-1T	*Rana temporaria*	GLLSGLKKVGKHVAKNVAVSLMDSLKCKISGDC	Hh: 36.36% Ee: 12.12% Cc: 51.52%	33	40 (*E. coli*)	[66]
					120 (*S. aureus*)	
					150 (*C. albicans*)	
Ranatuerin-2AUa	*Rana aurora aurora*	GILSSFKGVAKGVAKNLAGKLLDELKCKITGC	Hh: 53.12% Ee: 9.38% Cc: 37.50%	32	5 (*E. coli*)	[59]
					5 (*P. aeruginosa*)	
					10 (*K. pneumoniae*)	
					5 (*E. cloacae*)	
					>40 (*P. mirabilis*)	
					20 (*S. aureus*)	
					20 (*S. epidermidis*)	
					5 (*Streptococcus* Group B)	
					>40 (*E. faecalis*)	
Ranatuerin-2BYa	*Rana boylii*	ILSTFKGLAKGVAKDLAGNLLDKFKCKITGC	Hh: 48.39% Ee: 19.35% Cc: 32.26%	31	7 (*E. coli*) 27 (*S. aureus*) NA (*C. albicans*)	[50]
Ranatuerin-2BYb		IMDSVKGLAKNLAGKLLDSLKCKITGC	Hh: 44.44% Ee: 11.11% Cc: 44.44%	27	17 (*E. coli*) NA (*S. aureus*) NA (*C. albicans*)	
Ranatuerin-2Cb	*Rana clamitans*	GLFLDTLKGLAGKLLQGLKCIKAGCKP	Hh: 44.44% Ee: 7.41% Cc: 48.15%	27	2 (*E. coli*) 40 (*S. aureus*) 46 (*C. albicans*)	[65]
Ranatuerin-2CSa	*Rana cascadae*	GILSSFKGVAKGVAKDLAG KLLETLKCKITGC	Hh: 46.88% Ee: 18.75% Cc: 34.38%	32	5 (*E. coli*) 10 (*S. aureus*)	[67]
Ranatuerin-2Pb	*Rana pipiens*	SFLTTVKKLVTNLAALAGTVIDTIKCKVTGGCRT	Hh: 35.29% Ee: 32.35% Cc: 32.35%	34	8 (*E. coli*) >256 (*P. aeruginosa*) 8 (*S. aureus*) 16 (MRSA) >512 (*E. faecalis*) 8 (*C. albicans*)	[68]
Ranatuerin-2YJ	*Rana* *dybowskii*	GLMDIFKVAVNKLLAAGMNKPRCKAAHC	Hh: 39.29% Ee: 17.86% Cc: 42.86%	28	22.5 (*E. coli*) 22.5 (*S. aureus*)	[69]

**Table 4 toxins-14-00722-t004:** The origin, primary structure, secondary structure, length, and antibacterial activity of ranacyclins. Secondary structure prediction was performed by GOR IV algorithm (https://npsa-prabi.ibcp.fr/cgi-bin/npsa_automat.pl?page=/NPSA/npsa_gor4.html accessed on 27 September 2022); Hh: Alpha helix; Ee: Extended strand; Cc: Random coil; MIC: Minimum inhibitory concentrations (μM); NA: not active.

Peptides Name	Species	Primary Structure	Secondary Structure	Length	MIC (μM)	Ref.
Ranacyclin-B-RN1	*Hylarana* *nigrovittata*	SALVGCWTKSYPPKPCFGR	Ee: 10.53% Cc: 89.47%	19	6 (*S. aureus*)	[16]
Ranacyclin-B-RN2		SALVGCGTKSYPPKPCFGR	Ee: 10.53% Cc: 89.47%	19	12.7 (*S. aureus*)	
Ranacyclin E	*Rana temporaria*	SAPRGCWTKSYPPKPCK	Ee: 35.29% Cc: 64.71%	17	NA (*E. coli* D21) 9 (*Y. pseudotuberculosis* YP III) 80 (*P. syringae pv tabaci*) 3 (*B. megaterium* Bm11) 7 (*S. lentus*) 5 (*M. luteus*) NA (*C. albicans ATCC 10231*) 7.4 (*C. tropicalis*) 3.4 (*C. guiller-mondii*) 32 (*P. nicotianae spores*)	[70]
Ranacyclin T	*Rana esculenta*	GALRGCWTKSYPPKPCK	Ee: 29.41% Cc: 70.59%	17	30 (*E. coli* D21) 5 (*Y. pseudotuberculosis* YP III) 16 (*P. syringae pv tabaci*) 3 (*B. megaterium* Bm11) 10 (*S. lentus*) 8 (*M. luteus*) 22 (*C. albicans ATCC 10231*) 14 (*C. tropicalis*) 1 (*C. guiller-mondii*) 16 (*P. nicotianae spores*)	
Ranacyclin-NF	*Pelophylax nigromaculatus*	GAPRGCWTKSYPPQPCF	Ee: 23.53% Cc: 76.47%	17	>512 (*E. coli*) >512 (*P. aeruginosa*) >512 (*K. pneumoniae*) >512 (*S. aureus*) >512 (*MRSA*) >512 (*E. faecalis*)	[72]
Ranacyclin-NF1		GAPRGCWTKSYPPQPCF	Ee: 23.53% Cc: 76.47%	17	>512 (*E. coli*) >512 (*P. aeruginosa*) >512 (*K. pneumoniae*) >512 (*S. aureus*) >512 (*MRSA*) >512 (*E. faecalis*)	
Ranacyclin-NF3L		GALRGCWTKSYPPQPCF	Ee: 23.53% Cc: 76.47%	17	>512 (*E. coli*) >512 (*P. aeruginosa*) >512 (*K. pneumoniae*) >512 (*S. aureus*) >512 (*MRSA*) >512 (*E. faecalis*)	

**Table 5 toxins-14-00722-t005:** The origin, primary structure, secondary structure, length, and antibacterial activity of temporins. Secondary structure prediction was performed by GOR IV algorithm (https://npsa-prabi.ibcp.fr/cgi-bin/npsa_automat.pl?page=/NPSA/npsa_gor4.html accessed on 27 September 2022); Hh: Alpha helix; Ee: Extended strand; Cc: Random coil; MIC: Minimum inhibitory concentrations (μM); NA: not active.

Peptides Name	Species	Primary Structure	Secondary Structure	Length	MIC (μM)	Ref.
Temporin-1BYa	*Rana boylii*	FLPIIAKVLSGLL	Ee: 61.54% Cc: 38.46%	13	NA (*E. coli*) 15 (*S. aureus*) NA (*C. albicans*)	[50]
Temporin-1Gb	*Rana grylio*	SILPTIVSFLSKFL	Ee: 42.86% Cc: 57.14%	14	NA (*E. coli*) 24 (*S. aureus*) NA (*C. albicans*)	[76]
Temporin-1Gc		SILPTIVSFLTKFL	Ee: 57.14% Cc: 42.86%	14	NA (*E. coli*) 25 (*S. aureus*) NA (*C. albicans*)	
Temporin-1Gd		FILPLIASFLSKFL	Ee: 14.29% Cc: 85.71%	14	NA (*E. coli*) 12 (*S. aureus*) NA (*C. albicans*)	
Temporin-LT1	*Hylarana* *latouchii*	FLPGLIAGIAKML	Ee: 38.46% Cc: 61.54%	13	NA (*P. fluorescens*) 12.5 (*S. aureus*) 25 (*B. subtilis*) NA (*C. albicans*)	[88]
Temporin-LT2		FLPIALKALGSIFPKIL	Ee: 29.41% Cc: 70.59%	17	NA (*P. fluorescens*) 12.5 (*S. aureus*) 50 (*B. subtilis*) NA (*C. albicans*)	
Temporin-GHaR	*Hylarana guentheri*	FLQRIIGALGRLF	Ee: 61.54% Cc: 38.46%	13	6.2 (*E. coli*) 12.5 (*E. coli* D31) 6.2 (*P. aeruginosa*) 25 (*P. aeruginosa* PAO1) 1.6 (*S. aureus*) 12.5 (*B. subtilis*) 3.1 (*S. mutans*) 3.1 (*MRSA*) 12.5 (*C. albicans*)	[89]
Temporin-GHaR6R		FLQRIRGALGRLF	Ee: 53.85% Cc: 46.15%	13	12.5 (*E. coli*) 25 (*E. coli* D31) >50 (*P. aeruginosa*) 3.1 (*P. aeruginosa* PAO1) 3.1 (*S. aureus*) 25 (*B. subtilis*) 12.5 (*S. mutans*) 6.2 (*MRSA*) 50 (*C. albicans*)	
Temporin-GHaR7R		FLQRIIRALGRLF	Ee: 53.85% Cc: 46.15%	13	3.1 (*E. coli*) 12.5 (*E. coli* D31) >50 (*P. aeruginosa*) >50 (*P. aeruginosa* PAO1) 3.1 (*S. aureus*) >50 (*B. subtilis*) 6.2 (*S. mutans*) 3.1 (*MRSA*) 25 (*C. albicans*)	
Temporin-GHaR8R		FLQRIIGRLGRLF	Ee: 61.54% Cc: 38.46%	13	3.1 (*E. coli*) 12.5 (*E. coli* D31) 50 (*P. aeruginosa*) 6.2 (*P. aeruginosa* PAO1) 1.6–3.1 (*S. aureus*) 12.5 (*B. subtilis*) 6.2 (*S. mutans*) 3.1 (*MRSA*) 12.5 (*C. albicans*)	
Temporin-GHaR9R		FLQRIIGARGRLF	Ee: 53.85% Cc: 46.15%	13	>50 (*E. coli*) >50 (*E. coli* D31) >50 (*P. aeruginosa*) >50 (*P. aeruginosa* PAO1) 6.2 (*S. aureus*) >50 (*B. subtilis*) >50 (*S. mutans*) >50 (*MRSA*) >50 (*C. albicans*)	
Temporin-GHaR9W		FLQRIIGAWGRLF	Ee: 53.85% Cc: 46.15%	13	12.5 (*E. coli*) 12.5 (*E. coli* D31) >50 (*P. aeruginosa*) 25 (*P. aeruginosa* PAO1) 3.1 (*S. aureus*) 12.5 (*B. subtilis*) 6.2 (*S. mutans*) 6.2 (*MRSA*) 25 (*C. albicans*)	

**Table 7 toxins-14-00722-t007:** The origin, primary structure, secondary structure, length, and antibacterial activity of dybowskins. Secondary structure prediction was performed by GOR IV algorithm (https://npsa-prabi.ibcp.fr/cgi-bin/npsa_automat.pl?page=/NPSA/npsa_gor4.html accessed on 27 September 2022); Hh: Alpha helix; Ee: Extended strand; Cc: Random coil; MIC: Minimum inhibitory concentrations (μM); NA: not active.

Peptides Name	Species	Primary Structure	Secondary Structure	Length	MIC (μM)	Ref.
Dybowskin-1	*Rana dybowskii*	FLIGMTHGLICLISRKC	Ee: 47.06% Cc: 52.94%	17	>52.5 (*E. coli*) >52.5 (*K. pneumoniae*) >52.5 (*P. mirabilis*) >52.5 (*P. aeruginosa*) 13.1 (*S. aureus*) 26.3 (*B. subtilis*) 31.5 (*S. epidermidis*) >52.5 (*S. dysenteriae*) 6.6 (*M. luteus*) >52.5 (*C. albicans*)	[99]
Dybowskin-2		FLIGMTQGLICLITRKC	Ee: 47.06% Cc: 52.94%	17	31.4 (*E. coli*) 31.4 (*K. pneumoniae*) >52.4 (*P. mirabilis*) >52.4 (*P. aeruginosa*) 7.9 (*S. aureus*) 13.1 (*B. subtilis*) 13.1 (*S. epidermidis*) 26.2 (*S. dysenteriae*) 3.3 (*M. luteus*) 52.4 (*C. albicans*)	
Dybowskin-3		GLFDVVKGVLKGVGKNVAGSLLEQLKCKLSGGC	Hh: 27.27% Ee: 39.39% Cc: 33.33%	33	4.5 (*E. coli*) 4.5 (*K. pneumoniae*) >30.2 (*P. mirabilis*) >30.2 (*P. aeruginosa*) 9 (*S. aureus*) 15.1 (*B. subtilis*) 18.1 (*S. epidermidis*) 18.1 (*S. dysenteriae*) 1.9 (*M. luteus*) 30.2 (*C. albicans*)	
Dybowskin-4		VWPLGLVICKALKIC	Ee: 33.33% Cc: 66.67%	15	9.1 (*E. coli*) 9.1 (*K. pneumoniae*) >60.4 (*P. mirabilis*) >60.4 (*P. aeruginosa*) 1.9 (*S. aureus*) 3.8 (*B. subtilis*) 3.8 (*S. epidermidis*) 18.1 (*S. dysenteriae*) 0.9 (*M. luteus*) 15.1 (*C. albicans*)	
Dybowskin-5		GLFSVVTGVLKAVGKNVAKNVGGSLLEQLKCKISGGC	Hh: 24.32% Ee: 35.14% Cc: 40.54%	37	16.8 (*E. coli*) 13.6 (*K. pneumoniae*) >27.2 (*P. mirabilis*) >27.2 (*P. aeruginosa*) 6.8 (*S. aureus*) 13.6 (*B. subtilis*) 13.6 (*S. epidermidis*) 13.6 (S. dysenteriae) 1.8 (*M. luteus*) 13.6 (*C. albicans*)	
Dybowskin-6		FLPLLLAGLPLKLCFLFKKC	Ee: 60.00% Cc: 40.00%	20	>44 (*E. coli*) >44 (*K. pneumoniae*) >44 (*P. mirabilis*) >44 (*P. aeruginosa*) 5.5 (*S. aureus*) 22 (*B. subtilis*) 22 (*S. epidermidis*) >44 (*S. dysenteriae*) 2.7 (*M. luteus*) 22 (*C. albicans*)	
Dybowskin-1CDYa	*Rana dybowskii*	IIPLPLGYFAKKT	Ee: 15.38% Cc: 84.62%	13	3 (*E. coli*) 6 (*S. aureus*)	[100]
Dybowskin-2CDYa		SAVGRHGRRFGLRKHRKH	Ee: 27.78% Cc: 72.22%	18	3 (*E. coli*) 6 (*S. aureus*)	

**Table 8 toxins-14-00722-t008:** The origin, primary structure, secondary structure, and length of antioxidant peptides. Secondary structure prediction was performed by GOR IV algorithm (https://npsa-prabi.ibcp.fr/cgi-bin/npsa_automat.pl?page=/NPSA/npsa_gor4.html accessed on 27 September 2022); Hh: Alpha helix; Ee: Extended strand; Cc: Random coil.

Peptides Name	Species	Primary Structure	Secondary Structure	Length	Ref.
Andersonin-C1	*Odorrana margaratae*	TSRCIFYRRKKCS	Ee: 53.85% Cc: 46.15%	13	[105]
Andersonin-G1	*Odorrana andersonii*	KEKLKLKAKAP KCYNDKLACT	Ee: 23.81% Cc: 76.19%	21	
Andersonin-H3	*Odorrana margaratae*	VAIYGRDDRSDVCR QVQHNWLVCDTY	Ee: 42.31% Cc: 57.69%	26	
Antioxidin-RP1	*Rana pleuraden*	AMRLTYNKPCLYGT	Ee: 28.57% Cc: 71.43%	14	[101,105]
Cathelicidin-OA1	*Odorrana andersonii*	IGRDPTWSHLAASCLKCIFDDLPKTHN	Hh: 29.63% Ee: 7.41% Cc: 62.96%	27	[106]
Nigroain-B-MS1	*Hylarana maosuoensis*	CVVSSGWKWNY KIRCKLTGNC	Ee: 47.62% Cc: 52.38%	21	[107]
OA-VI12	*Odorrana andersonii*	VIPFLACRPLGL	Ee: 50.00% Cc: 50.00%	12	[104]
OA-GL21	*Odorrana andersonii*	GLLSGHYGRVVSTQSGHYGRG	Ee: 52.38% Cc: 47.62%	21	[108]
OM-LV20	*Odorrana margaretae*	LVGKLLKGAVGDVCGLLPIC	Ee: 45.00% Cc: 55.00%	20	[109]
OM-GF17	*Odorrana margaretae*	GFFKWHPRCGEEHSMWT	Ee: 35.29% Cc: 64.71%	17	[110]
OS-LL11	*Odorrana schmackeri*	LLPPWLCPRNK	Cc: 100.00%	11	[103]
Pleurain-A1	*Rana pleuraden*	SIITMTKEAKLPQLW KQIACRLYNTC	Hh: 19.23% Ee: 30.77% Cc: 50.00%	26	[101]
Pleurain-D1	*Rana pleuraden*	FLSGILKLAFKIP SVLCAVLKNC	Ee: 47.83% Cc: 52.17%	23	
Pleurain-E1	*Rana pleuraden*	AKAWGIPPHVIPQI VPVRIRPLCGNV	Ee: 30.77% Cc: 69.23%	26	
Salamandrin-I	*Salamandra salamandra*	FAVWGCADYRGY	Ee: 58.33% Cc: 41.67%	12	[111]

## Data Availability

Not applicable.

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
