# Peer review of "Peptides Isolated from Amphibian Skin Secretions with Emphasis on Antimicrobial Peptides"

_toxins, 2022, doi:10.3390/toxins14100722_

Round 1

Reviewer 1 Report

The skin of amphibians contains hundreds of peptides with biological functions including antimicrobial peptides, antioxidant peptides, bradykinins, insulin-releasing peptides, and other peptides. The authors of this review have collected information on the origin, primary structure and functions of these skin peptides to foster application and further studies on amphibian bioactive peptides.

Prior to publication, the following details should be considered by the authors:

1)     Table 1: give all bacterial names in italics

2)     Lane 79: give citation without first name initials of first author

3)     Lane 80: write “did not” rather than “didn’t”

4)     Lane 97: write “In their study, …”

5)     Table 3: give all bacterial names in italics

6)     Table 8: Write “Odorrana andersonii” at Ref #90

7)     Table 8: Write “Odorrana andersonii” at Ref #94

8)     Lane 235: give citation without first name initials of first author

9)     Lane 242: give citation without first name initials of first author

10)  List of References: Numbering of references appears double

Author Response

Dear Reviewer,

Thank you so much for your timely review for our manuscript. We appreciate you for the comments concerning our manuscript entitled “A Review on Peptides Isolated from Amphibians Skin Secretions” (toxins-1908518). The comments are all valuable and very helpful for revising and improving our paper, and give us the important guiding significance to our researches. We have studied the comments carefully and have made corrections which we hope to meet with approval. Revised portion are marked in red in the paper. Our responses are provided in the document "Reviewer 1". Please see the attachment.

Reviewer 2 Report

Antimicrobial peptides or AMPs are vital leads for the further development of antibiotics against MDR pathogens. Amphibian skins are a rich source of various AMPs. Review articles have been written on AMPs isolated and characterized from amphibian skins. The current manuscript needs to be improved, as it lacks a strong focus. Temporins are group of AMPs which are less toxic to human cells but active against bacteria. Mode of action, engineering and improved activity of temporins are well documented, some examples are Cell Mol Life Sci. 2011 68(13):2267-80. doi: 10.1007/s00018-011-0718-2, Biochim Biophys Acta. 2009, 1788(8):1610-9. doi: 10.1016/j.bbamem.2009.04.021, PLoS One. 2013 ;8(9):e72718. doi: 10.1371/journal.pone.0072718, J Biol Chem. 2011 286(27):24394-406. doi: 10.1074/jbc.M110.189662, Int J Mol Sci. 2020 21(16):5773. doi: 10.3390/ijms21165773. Current manuscript has no figures. I suggest that authors must include relevant figures to illustrate mode of action of the peptides.

Author Response

Dear Reviewer,

Thank you so much for your timely review for our manuscript. We appreciate you for the comments concerning our manuscript entitled “A Review on Peptides Isolated from Amphibians Skin Secretions” (toxins-1908518). The comments are all valuable and very helpful for revising and improving our paper, and give us the important guiding significance to our researches. We have studied the comments carefully and have made corrections which we hope to meet with approval. Revised portion are marked in red in the paper. Our responses are provided in the document "Reviewer 2". Please see the attachment.

Reviewer 3 Report

The title and abstract of this manuscript promise a review of peptides isolated from amphibian skin secretions that exhibit a broad range of activities. In fact, the presented material is mainly focused on antimicrobial peptides, and although the collection of tabulated characteristics possibly offers a useful reference for some readers, the overview is entirely descriptive and lacks the synthesis and critical evaluation that would make the paper interesting to a wider audience. For instance, the manuscript does not sufficiently consider the relation between the structure of the various peptides and their antimicrobial activity and possible biological toxicity. More broadly, it seems worthwhile to discuss the potential of amphibian skin as a source for the discovery of antimicrobial peptides that have novel modes of action, and thus may help address the concerning rise in antimicrobial resistance. Indeed, does any of the listed peptides belong to an unusual class of antibiotics? Peptides with other activity (antioxidant, insulinotropic and others) are described in minimal detail. The writing needs attention throughout to remove ambiguities and increase clarity of expression.

Author Response

Dear Reviewer,

Thank you so much for your timely review for our manuscript. We appreciate you for the comments concerning our manuscript entitled “A Review on Peptides Isolated from Amphibians Skin Secretions” (toxins-1908518). The comments are all valuable and very helpful for revising and improving our paper, and give us the important guiding significance to our researches. We have studied the comments carefully and have made corrections which we hope to meet with approval. Revised portion are marked in red in the paper. Our responses are provided in the document "Reviewer 3". Please see the attachment.

Round 2

Reviewer 2 Report

Comments are addressed. Manuscript is now suitable for publication. 

Reviewer 3 Report

No further comments